# Trans*Forming Access and Care in Rural Areas: A Community-Engaged Approach

**DOI:** 10.3390/ijerph182312700

**Published:** 2021-12-02

**Authors:** Megan E. Gandy, Kacie M. Kidd, James Weiss, Judith Leitch, Xavier Hersom

**Affiliations:** 1School of Social Work, West Virginia University, 29 Beechurst Ave, Morgantown, WV 26505, USA; jaweiss@mix.wvu.edu; 2Division of Adolescent Medicine, Department of Pediatrics, Medical Director, Gender & Sexual Development Clinic, WVU Medicine Children’s, West Virginia University School of Medicine, 1 Medical Center Drive, Morgantown, WV 26506, USA; kacie.kidd@hsc.wvu.edu; 3Department of Social Work, University of North Carolina at Greensboro, 1400 Spring Garden Street, Greensboro, NC 27412, USA; jeleitch@uncg.edu; 4Transforming Healthcare Community Advisory Board Member, School of Social Work, West Virginia University, 29 Beechurst Ave, Morgantown, WV 26505, USA; xavierhersomcabmember@gmail.com

**Keywords:** transgender, nonbinary, transsexual, health, health care, access, qualitative, mixed methods, CBPR, discrimination

## Abstract

Research indicates that rural transgender and gender diverse (TGD) populations have a greater need for health services when compared with their urban counterparts, face unique barriers to accessing services, and have health disparities that are less researched than urban TGD populations. Therefore, the primary aim of this mixed-methods study (*n* = 24) was to increase research on the health care needs of TGD people in a rural Appalachian American context. This study was guided by a community-engaged model utilizing a community advisory board of TGD people and supportive parents of TGD children. Quantitative results indicate that travel burden is high, affirming provider availability is low, and the impacts on the health and mental health of TGD people in this sample are notable. Qualitative results provide recommendations for providers and health care systems to better serve this population. Integrated mixed-methods results further illustrate ways that rural TGD people and families adapt to the services available to them, sometimes at significant economic and emotional costs. This study contributes to the small but growing body of literature on the unique needs of rural TGD populations, including both adults and minors with supportive parents, by offering insights into strategies to address known disparities.

## 1. Introduction

Transgender and gender diverse (e.g., nonbinary, genderqueer, agender, demigender, etc.) (TGD) individuals are those whose gender identity and sex assigned at birth do not align. This population experiences significant marginalization and health inequities through numerous factors including social stigma, discrimination, harassment, and abuse [1]. These inequities are compounded by distrust for health providers due to personal and/or historical experiences of discrimination in health settings that result in avoidance of necessary medical and mental health care, including life-saving and preventative health care [2,3]. In a national survey of TGD adults in the United States, 33% had had a negative experience with a provider related to their gender identity in the last year, and nearly one in four had avoided seeing a doctor in the previous year when they needed to out of fear of being mistreated [4]. In addition to historical and experienced discrimination in medical settings, TGD individuals face numerous additional barriers to accessing medical care. This is especially true for gender-affirming medical care [4]. This care, which is inclusive of mental health support as well as gender-affirming medical and surgical interventions, is limited by cost, as many insurance options limit or deny coverage for gender-affirming interventions, and distance, as well-educated providers of gender-affirming care can be hard to find. For example, one study found that while 78% of TGD respondents wanted hormone therapy, only 49% had ever received it [4]. 

Barriers to care are further heightened for TGD minors (age < 18 years) who experience even more marked health inequities, including up to four times the rate of suicidal ideation of cisgender (non-TGD) peers [5]. These barriers include TGD minors and their families having difficulty identifying a knowledgeable provider with training in pediatric gender-affirming care as well as uncoordinated care and significant insurance limitations [6,7]. Additionally, this care has come under legislative threat, as numerous states have considered legislation that would ban gender-affirming medical and surgical interventions for TGD minors, which would create even more barriers to receiving this needed care [8,9].

Rural Americans also face health disparities with higher rates of heart disease, cancer, infant mortality, and a host of other medical ailments than those living in more urban settings [10,11,12,13]. The source of these disparities is multifactorial and associated with reduced or lack of access to medical care, as well as socioeconomic factors, including differences in substance use patterns and lower income and educational attainment [14,15,16]. When comparing rural and urban transgender health care needs, rural transgender persons have significantly higher needs in mental health, substance use, and health issues stemming from sexual risk behaviors when compared with their urban counterparts [17]. This is particularly fundamental for TGD individuals living in states that lack transgender-specific policies that provide protection or prohibit discrimination, as is the case in West Virginia [18]. Transgender individuals who live in states without these policies are much more likely to avoid health care out of fear of discrimination [19].

Rural TGD individuals are a markedly understudied population, particularly those living in the Appalachian geographic regions, and those with nonbinary or gender-expansive identities [20]. This is a significant gap in the literature, but preliminary research suggests there is an association between geographic location and anxiety and depression in transgender individuals [21]. The few existing studies suggest that this population faces compounding social stigma and increased barriers to all health care, but especially gender-affirming medical and surgical care [17,20,22,23,24]. Given the dearth of research exploring the facilitators and barriers to health care access for rural TGD people [25], exploratory mixed-methods approaches are needed. Additionally, the lack of representation of rural TGD Americans in the larger national conversation about TGD health care belabors the need for stakeholder-engaged research to elevate this often-unseen population. 

To address these gaps in the literature, this study aimed to use a stakeholder-engaged mixed-methods approach to begin to determine what TGD people, both adults and minors, require in a rural Appalachian American context such as West Virginia to have their health care needs met. 

## 2. Methods

This mixed-methods study examined the following research question: What do TGD people need from medical providers in order to have their health care needs met in West Virginia? The study was approved by the first author’s institutional review board for human subjects. Data were collected from May to September 2021. This study used a convergent mixed-methods design, in which the study design, setting, and participants were first established, then the quantitative data were collected and analyzed, followed by the same for the qualitative data; then, the findings were integrated into analysis and discussion [26].

### 2.1. Community Engagement

Classification of community engagement in research can be considered on a spectrum based on the amount of engagement, communication, and decision-making power held by community members in a project [27]. Those factors delineate categories in the Community Engagement Continuum presented by McCloskey et al., spanning five categories ranging from “outreach” to “shared leadership” [28]. It is important to note that not all research that calls itself “community engaged” is engaging with the community in the same ways. Studies that gather input from community stakeholders but do not include them in decision-making processes, have bidirectional communication, or explicitly address power and privilege do not advance equity and justice efforts in the same ways as studies that accomplish those objectives. 

This study’s approach falls in the fourth out of the five categories in the Community Engagement Continuum, “collaborate”, which is the second from the top in terms of the level of community engagement, because of its shared decision making, bidirectional communication, and partnership with community on each aspect of project development and implementation [28]. Researchers of this study engaged a community advisory board of TGD adults and parents of TGD minors who live in West Virginia throughout the research process, including the design, data collection, and analysis of results. Meetings were held monthly throughout the project. Members held the responsibility of deciding on survey and focus group questions, as well as identifying outlets for recruitment. Members were invited to become research associates by learning how to conduct focus groups, becoming trained in the Ethical Conduct for Human Subjects Research program, and being listed as research personnel in the IRB protocol. Members were paid for their efforts and included in the authorship of results, thus recognizing their expertise. This process is described in more detail elsewhere [29].

Community- and stakeholder-engaged research with both TGD people and parents of TGD minors is important because historical and modern marginalization of gender diverse voices, particularly those of Black, Indigenous, and other people of color, as well as those from rural areas, has profoundly impeded the recognition that these communities need to meet their health care needs [30,31]. Therefore, this research was performed in collaboration with the TGD community instead of engaging them from the outside as merely a subject to be studied, with the goal of increasing the validity of the results [30]. 

### 2.2. Procedures

This project was planned prior to the onset of the COVID-19 pandemic but not implemented until after the pandemic settled into the region. Thus, the plans for a typical in-person, community-engaged research project were adjusted due to the need to be socially distanced, remain home, and avoid the spread of the virus. Recruitment was conducted online via social networks related to TGD communities in West Virginia. Eligibility criteria were as follows: over age 18, or under age 18 with parental consent and participation; self-identification as transgender or gender diverse (or having a minor child who identifies); English-speaking; lives currently in West Virginia or have lived there in the past 12 months; internet access. Parents of TGD minors were eligible to participate on behalf of their minor child. Participants completed an online survey and were invited to attend an online focus group. Focus groups were offered using both synchronous (Zoom) and asynchronous (www.focusgroupit.com (accessed on 7 July 2021)) options. Online focus groups are especially efficacious for research with TGD people due to a desire to remain anonymous while discussing sensitive information [32]. Asynchronous focus groups are held in a moderated forum-style discussion board that followed the same interview guide as the synchronous sessions. Participants could read and respond to other participant answers just like they would be able to in a synchronous session [33]. The asynchronous group was an option for those who were unable to attend one of the synchronous Zoom focus groups. Focus group participants received a USD 25 incentive for their participation. 

### 2.3. Sample 

In total, 24 individuals completed the online survey and participated in one of six online focus groups. For the study, 4 synchronous online video focus groups were attended by 16 participants, and 8 participants engaged in the asynchronous online focus group. A total of 18 participants in this sample were over age 18, and 6 participants were minors (parents answered questions on behalf of their minor child). Demographics of the final sample are presented in Table 1. TGD participants’ age ranged from 6 to 62 years; the mean age was 28.71 (SD = 14.47). The sample was mostly non-Hispanic white (83.3%), transgender (87.5%), and mostly educated with a 4-year college degree (33.3%).

### 2.4. Online Survey and Quantitative Variables

Online survey questions were developed in collaboration with the community advisory board and focused on variables related to health care access, general health and mental health status, and demographic information. Variables for access to health care included travel distance by hours, travel out of state, transportation method, insurance coverage, and access to desired gender-affirming care. Provider-related variables included knowledge of the provider, gender-inclusive care, and avoidance of medical visits. Health and mental health variables included the DASS-21, one question asking overall health self-rating, and 12-month history of suicidal thoughts or attempts. The DASS-21 is a self-report measure of depression, anxiety, and stress with good reliability and validity with this population [34,35].

### 2.5. Online Focus Group and Qualitative Domains 

Online focus group questions were developed in collaboration with the Community Advisory Board and focused on illuminating the barriers and facilitators to accessing gender-inclusive health care for any medical need in West Virginia. Categories included health areas of need, positive and negative health care experiences, barriers to accessing health care, and provider competence in gender-inclusive care. Focus group interview questions are included in Table 2. 

### 2.6. Analysis 

Quantitative data from the online survey for the 24 focus group participants were cleaned and prepared for analysis using IBM SPSS version 28 for Windows. No missing data were present in this subsample of the study. Bivariate relationships were explored to determine if there was any statistically significant relationship between access to care and the mental health of participants. Nonparametric tests were used (Kruskal–Wallis *H* for bivariate pairs where the independent variable was continuous, and the dependent variable was categorical) due to the small sample size and nonrepresentative nature of the data. Qualitative data were analyzed using content analysis by two separate coders who independently coded the data, then compared codes and memos until consensus was reached on the major categories of the data. Analysis was organized using Atlas-ti Cloud online qualitative software. Quotes were then pulled that best illustrated each category. Quantitative data were integrated with the qualitative findings to create a coherent understanding of the mixed-methods data.

## 3. Results

Findings will be presented following the convergent mixed-methods design: quantitative results, then qualitative results, and then integrated mixed-methods results will be discussed [26]. Findings will focus on the following research question: What do transgender and gender diverse people need from providers to have their health care needs met in West Virginia?

### 3.1. Quantitative Results

#### 3.1.1. Transportation

Overall, participants had to travel on average nearly an hour and a half (1.425 h) to access gender-related care. Further, 70% of participants had to travel out of state for gender-affirming care (or would have to if they have not yet sought out gender-affirming care). Participants were asked what transportation method they used to travel to the doctor for any health care reason. Although most participants relied on their own transportation (83.3%), three accepted a ride from someone else (12.5%), and one person relied on rideshare apps or public transportation (4.16%), in addition to accepting a ride from someone else.

#### 3.1.2. Gender-Affirming Care Access

Participants were asked whether they had health insurance and whether their insurance covered gender-related care. All but one participant had health insurance (95.8%), but insurance coverage for gender-related care varied, with 37.5% having coverage, 20.8% no coverage, and 37.5% not sure. Participants were asked if they had sought health care for gender-affirmation purposes. In total, 19 (79.2%) participants had sought gender-affirming care, 4 participants (16.7%) wanted or planned to but had not yet, and 1 person (4.2%) did not plan on seeking any health care for gender affirmation. Participants were asked whether they received the gender-affirming care that they desired. Overall, 10 participants (41.7%) had received the gender-affirming care they desired. Participants were then asked why they had not received the gender-affirming care they desired (if applicable). Cost was a significant barrier for some. Of the care received, five participants (20.8%) had received hormone replacement therapy and wished to have surgery but could not afford it. Two participants (8.3%) wished to have hormone replacement therapy or surgery but could not afford either. Two people (8.3%) had not received the gender-affirming care they desired because they did not know where to seek it. Other barriers to accessing care included lack of provider trust (*n* = 3, 12.5%), fear of discrimination in employment and housing (*n* = 1, 4.2%), and lack of parental approval (*n* = 1, 4.2%). 

#### 3.1.3. Perspectives on Providers 

Participants were asked whether they felt their providers were knowledgeable about medical terminology and procedures regarding gender-affirming care, regardless of whether they provided those services. Response options were in five-point Likert-style answers presented in Table 3. Most participants answered that their provider might or might not be knowledgeable. 

Participants were asked whether their primary care doctor (or whoever they see most often) provided gender-inclusive care (the definition provided to participants was that they would react positively to the person’s gender identity). Response options were in five-point Likert-style answers presented in Table 3. Most participants responded that they always provided gender-inclusive care. If their response was anything but “always”, they were asked why they responded in that way and could select all that apply (thus, percentages would add up to greater than 100%). Eleven (45.8%) participants said that the provider did not have enough knowledge about gender-related health care issues; six (25%) said that the provider did not address their gender-related health care needs, only other medical needs; three (12.5%) said that the office did not provide a welcoming environment or did not use inclusive forms; one (4.2%) participant said that their provider was not comfortable with gender-diverse patients; one (4.2%) participant said they were not out to their primary care physician, so they were constantly misgendered.

#### 3.1.4. Avoidance of Care

Participants were asked if they ever had a health care need but avoided visiting the doctor because of fear of discrimination. Half of the respondents (*n* = 12) stated that they had avoided receiving needed medical treatment in the past year out of fear of discrimination. Participants who said they had avoided care were then asked to identify what type of care they needed when they avoided a medical visit. The types of health care they needed or locations they sought care at included urgent care, care due to being sick, and a routine check-up visit but did not include emergency room care.

#### 3.1.5. Self-Reported Mental and Physical Health

Participants were asked to self-rate their overall health on a five-point Likert-style scale. Overall, participants’ reported physical health was average or better. Participants rated their overall health as follows: four (16.7%) excellent, seven (29.2%) good, nine (37.5%) average, and four (16.7%) poor. Participants were then asked to answer the DASS-21 scale (parents filling out the survey on behalf of their minor child were instructed to have the minor child fill out the DASS-21). Results are reported by subscale and severity category in Table 4. Those in the moderate-to-severe category in each subscale were depression over half (58.3%), anxiety nearly half (45.8%), and stress nearly half (45.8%). In the past 12 months, 9 (37.5%) participants had thought about suicide, 13 (54.2%) had not, and 2 (8.3%) did not wish to disclose. In the past 12 months, one (4.2%) person had attempted suicide.

#### 3.1.6. Bivariate Analyses

A Kruskal–Wallis *H* test was performed on each categorical independent variable relating to access to gender-affirming care, including travel time, travel out of state, insurance coverage, access to desired gender-affirming care, having a knowledgeable provider, having a gender-inclusive provider, and avoiding care due to fear of discrimination. The continuous dependent variables were the total score for each subscale of the DASS-21. In all 21 of the Kruskal–Wallis *H* tests on pairs of independent and dependent variables, there were no statistically significant differences in DASS-21 scores among groups in each categorical variable. For example, a Kruskal–Wallis *H* test was performed to detect the statistical significance of the mean difference in DASS-21 anxiety subscale scores among groups based on whether or not a provider was knowledgeable about gender-affirming health care (three groups formulated from the five-point Likert-style responses: definitely/probably yes, might or might not, definitely/probably no) and was found to be not statistically significant (*H*(3) = 4.890, *p* = 0.087). 

### 3.2. Qualitative Categories

Four categories were used to explore the data, including supportive health care experiences, harmful health care experiences, accessing health care needs, and perspectives on providers.

#### 3.2.1. Category One: Supportive Health Care Experiences

Overall, participants described an action as supportive if the provider gave them some sort of affirmation of their gender identity, but many also described it as positive if someone simply treated them with the same respect as any other patient. For example, one participant stated, “I had a general practitioner who actually listened to me and took care of my needs and didn’t treat me any differently than anybody else”. Participants also described neutral actions as supportive, such as when providers followed organizational policies, as one participant recalled, “a health care provider who was competent in working with my insurance to get my transition-related care covered appropriately without me having to do that work for them”.

#### 3.2.2. Category Two: Harmful Health Care Experiences 

In terms of describing harmful experiences, many participants talked about provider behavior that triggered gender dysphoria. One TGD participant relayed an experience they had in the emergency department with their transgender partner, “My husband… went in for abdominal pain. As soon as they found out that he was transgender, they immediately ordered a transvaginal ultrasound, which was completely, in our opinion, very inappropriate”. Participants also shared stories of how difficult it was to perform simple tasks related to their health care needs. For example, one participant shared, “I called to set up an appointment for an ultrasound on my genitalia. At this time, I had legally changed my name to a name that we would associate with women. When she asked what kind of ultrasound I answered ‘it says ultrasound of the scrotum’ and there was dead silence and… it took like 10 min, and she had to transfer me to somebody else for some reason to get this scheduled”. Similarly, other participants had difficulty just obtaining an appointment, such as one who shared, “as soon as I said my name, they would hang up on me… if I call to get a refill or need to schedule an appointment… just get hung up on. I have to constantly keep trying over and over and over and hope to get a different receptionist”.

Participants shared many stories of providers using deadnames and/or wrong pronouns despite the chart instructing them otherwise. When this happened, parents of TGD minors stepped in to advocate for their child’s needs, such as this participant who said, “I went up to them privately and I said I’d appreciate it when you’re calling us back to the room, if you can call us by this name and not the other, because you’re causing some really bad dysphoria”. This became a barrier when the child later did not want to go to the emergency room because of this experience. 

#### 3.2.3. Category Three: Accessing Health Care Needs 

Participants reflected on whether and when they avoid receiving health care, and how they find gender-affirming health care providers given the rural area and lack of providers. Some participants noted that they avoided their health care needs altogether for fear of discrimination by their providers. One participant remarked that they specifically avoided disclosing that they are transgender. They said, “I’ve never really felt safe talking to my providers about any of it. I just didn’t want to be discriminated against or dead named or anything like that, so I just didn’t tell them”. Another participant shared that they did not want to come out to their insurance company for fear that they would have difficulty gaining authorization for routine care. Participants also found some health care needs unavailable due to lack of insurance coverage or barriers to accessing coverage for gender-related care. One participant said, “considering I have been in transition for close to 20 years, that just seems absolutely ridiculous to me to spend a year doing [therapy] just to fulfill this insurance requirement to get something I need now”.

Participants were asked about how they seek out health care in WV and which factors influence how they seek care. Responses were coded according to networks, factors, and health care needs that they could not find. Many participants explained that they sought information on health care providers through TGD support groups. A significant barrier for some was the lack of health insurance coverage of gender-related care. A participant stated, “I need a more flexible or less gatekeeper kind of health insurance policy”. Still, participants shared that they lowered their expectations when they sought treatment, such as one participant who shared, “I have really low expectations. I put the ball on the ground, if not lower, and it just never fails to go pretty poorly”. TGD people likely face barriers due to intersectional identities such as race, as one participant put it, “there’s absolutely a race factor in there… overt or covert… that’s absolutely going to be a factor for me that may not be for the majority of people in this state”. This participant was one of only a handful of non-white participants in this study and is congruent with the small demographic of people of color in the state.

#### 3.2.4. Category Four: Perspective on Providers

Overall, many participants stated that providers needed more education regarding TGD health care issues and needs. One participant noted, “I feel like I’m constantly educating those that should be educated”. Additionally, participants explained that providers just need to treat TGD patients humanely like they would anyone else. One participant suggested, “The key kind of thing that may help people become more comfortable with trans people is just to experience trans people directly. Not in some sort of like a technical context…just seeing them as people first and not just like an anomaly as a patient”.

## 4. Discussion

Overall, this mixed-methods study offers unique insights into the facilitators and barriers to health care access for TGD people in a rural Appalachian American context.

### 4.1. Integrated Mixed-Methods Discussion

The quantitative findings related to transportation (travel time and the need to travel out of state for gender-affirming care) were further illustrated by the qualitative findings related to accessing gender-inclusive care: Participants had difficulty accessing doctors with expertise in gender-affirming care, they were concerned about discrimination from providers that were located closer to them, and there was simply a lack of available providers who were willing to provide care to TGD patients. 

In regard to mental health, the quantitative finding that nearly half of participants scored in the moderate-or-severe range in the DASS-21 subscales was further explained in the qualitative findings. Participants described negative health care experiences that triggered gender dysphoria. Their DASS-21 scores, however, are not fully explained by those discrimination experiences. The bivariate analyses of the DASS-21 with these variables were not statistically significant (likely due to a lack of statistical power from the small sample size). Nonetheless, this is of clinical significance, especially when considering the finding that over one-third of the sample reported suicidal thoughts in the past 12 months. 

The quantitative finding that almost half of the sample avoided visiting a doctor out of fear of discrimination was illuminated by the qualitative findings that centered on the impact of not disclosing their gender identity or simply not visiting a doctor for care and using home or over-the-counter remedies instead. This impact was evident in the qualitative responses relating to the difficulty participants had in accessing gender-affirming care. There was a higher burden placed on these TGD patients who had to navigate often stressful experiences related to exploring whether a provider was knowledgeable, whether their treatment would be covered by insurance, and whether they would be subjected to experiences that triggered gender dysphoria. 

Several participants reported that they had difficulty obtaining insurance coverage for their care, which sometimes included general health care needs that were made more difficult to authorize simply due to their status as a TGD person. One person stated that they did not wish to disclose their TGD identity to their health insurance provider for this reason. This qualitative finding connected to the quantitative finding that although participants had health insurance, their gender-related care may not be covered.

### 4.2. Implications

While the body of literature regarding the health care needs and experiences of TGD people living in rural areas is small, our findings are corroborated by other studies that suggest TGD people have difficulty accessing gender-affirming care and often avoid or delay accessing care altogether for fear of facing discrimination from health care professionals [36]. Additionally, one study pointed to the fact that many TGD people rely on TGD networks to seek out affirming care [37]. In contrast to studies conducted with TGD people living in urban areas, rural TGD people may have even greater difficulty accessing these networks of other TGD people due to geographic isolation [38]. Generally, our findings are consistent with those found in urban areas insofar as difficulty accessing gender-affirming care, socioeconomic marginalization, and lack of provider education on TGD issues remain as barriers to accessing health care services [39,40]. 

The results of this project have significant implications for practitioners to deliver gender-affirming care to their TGD people in rural areas. A recurring theme that arose from the data analysis is provider education regarding TGD issues. TGD people do not want the burden of educating their providers, which aligns with research on this topic [41,42], especially for non-metropolitan TGD people [43]. Additionally, our findings point to the need for providers to treat TGD people with the same dignity and respect as any other person. This finding is likely linked to the concepts of attitude toward or belief about TGD patients, which would suggest that providers should improve their education, but they should also improve their personal beliefs about TGD patients [44]. 

### 4.3. Limitations

The small, nonrepresentative sample limited generalizability and limited the scope of statistical analyses due to a lack of statistical power. Future studies should aim to collect data from a representative sample of rural TGD patients in Appalachia. This study only utilized online procedures for data collection due to the social distancing requirements of the COVID-19 pandemic response. Internet access is limited in rural areas. Therefore, this may have limited the participation of some otherwise eligible people. Furthermore, this study could not provide insight into the unique needs of pediatric patients, compared with adult patients. Future research is needed to better understand the unique needs of pediatric patients, compared with adult patients, as well as patients with state-based health insurance versus private insurance due to hindrances in authorization for gender-affirming treatment [45]. A limitation of this study related to pediatric patients was that, for the most part, parents responded to questions on behalf of their minor children. While most questions were appropriate for the parent to answer due to their role in consenting to health care treatment on behalf of their minor child, there would have been some opportunities for the minor to provide further detail. In some instances, the minor child was available and made this contribution; in other instances, the minor child was not available (due to age or developmental restrictions). Furthermore, the average age of this study’s sample was relatively low in comparison with the average age of the general West Virginia population [46]. Only one person over the age of 60 participated in this study; technology limitations may have contributed to this due to the online procedures used in this study. Future research involving rural TGD people should focus on the unique needs of the intersection between rural aging and TGD aging [47]. Finally, a limitation of this study is that it relied on self-report, and therefore, the results are subject to recall bias.

Despite these limitations, this study presents the first empirical documentation of the facilitators and barriers to health care access for TGD patients in a rural Appalachian American context. A strength of this study was its inclusion of both pediatric and adult TGD patients, representing a broader picture of health care access in this region. This study was further strengthened by its community-engaged, convergent, mixed-methods design, which was the first of its kind for Appalachian American TGD people.

## 5. Conclusions

The goal of this study was to determine how to make gender-inclusive health care more accessible to TGD people living in West Virginia. Given that little is known about the needs and experiences of this rural population, we utilized an exploratory, mixed-methods design to understand the facilitators and barriers to accessing health care services in this state. The results of this project provide health care professionals with key insights into the unique needs and experiences of TGD people living in rural areas and begin to highlight how health care services can be improved to provide comprehensive, affirming care. The unique needs of this rural population included issues related to geographic isolation, transportation burden due to low provider availability, and the economic costs of out-of-state travel for treatment access. Furthermore, there were emotional and physical costs related to avoiding going to the doctor for care, avoiding disclosure of gender identity out of fear of discrimination, and dealing with interactions that triggered gender dysphoria. Providers can improve their care to TGD patients by using chosen names and pronouns, seeking professional education around TGD care issues, being willing to engage in self-examination of one’s attitudes and beliefs about TGD people, and joining efforts to reduce barriers to health care access for this population.

## Figures and Tables

**Table 1 ijerph-18-12700-t001:** Demographics of final sample (*n* = 24).

Demographic	*n*	%
Education		
Elementary (currently) *	2	8.3
Middle (currently) *	1	4.2
High School (currently) *	2	8.3
High School/GED	4	16.6
2-year Degree	4	16.6
4-year Degree	8	33.3
Master’s Degree	2	8.3
PhD/MD Degree	1	4.2
Race		
White	20	83.3
White and Hispanic	1	4.2
White and Native Indian/Alaska Native	1	4.2
Black	1	4.2
Hispanic and Polish	1	4.2
Ethnicity		
Hispanic	4	16.6
Non-Hispanic	20	83.3
Gender Identity		
Nonbinary, agender, or genderqueer	7	29.2
Transgender (not otherwise specified)	7	29.2
Transgender woman	4	16.7
Transgender man	4	16.7
Transgender woman, nonbinary	1	4.2
Transgender man, nonbinary	1	4.2

* These are the grade levels of the TGD minors—parents answered this survey question on their behalf.

**Table 2 ijerph-18-12700-t002:** Structured focus group interview questions.

When is the last time you/your child went to a medical provider for any kind of treatment (related or unrelated to your transition, if applicable)?
2. In your view, what are your/your child’s most pressing health care needs? (For example, diabetes care, preventative care, trans-affirming care, etc.)
3. Can you think of a health care provider who did a great job providing care for you/your child? Prompt if yes: What about them created a positive experience for you?
4. Have you/your child ever had something happen during your medical visits that caused you/your child so much distress as a transgender or gender diverse person that you wanted not to return? Prompt if yes: Can you tell a story that illustrates what that experience was like?
5. When you/your child go to the doctor, do you/your child have a negative experience from the nonclinical staff, such as the receptionist, front-desk staff, medical records personnel, etc.? Prompt if yes: Do these interactions make you/your child hesitant to return to that doctor for your medical needs?
6. Has there been a time that you/your child needed medical care (for any reason) but did not receive it? (For example, too far to drive, cannot find a good doctor, cannot afford it, cannot have time off work, concerned about discrimination, etc.) Prompt if yes: Why did you/your child not receive it? Prompt if yes: What would you/your child need in order to receive the medical care next time you needed it?
7. Did you seek out specific providers to meet your/your child’s health care needs as a trans or gender diverse person? Prompt if yes: How did you locate resources, information, and health care providers given that you are in a rural area?
8. What is one thing that you think medical providers in West Virginia need to know or accomplish, in order to best serve you/your child’s needs as a transgender or gender diverse patient?

**Table 3 ijerph-18-12700-t003:** Results of questions about perspective on providers.

Do you feel that your health care providers are knowledgeable about medical terminology and procedures regarding gender-affirming care, regardless of whether they provide these services?		
Response	*n*	%
Definitely yes	2	8.3
Probably yes	4	16.7
Might or might not	11	45.8
Probably not	5	20.8
Definitely not	2	8.3
Thinking about your primary care doctor (or whoever you see most regularly), do you feel that they provide gender-inclusive care (meaning they react positively about your gender identity)?		
Response	*n*	%
Always	9	37.5
Often	5	20.8
Sometimes	6	25.0
Rarely	3	12.5
Never	1	4.2

**Table 4 ijerph-18-12700-t004:** DASS-21 results.

Subscale and Severity Category (Possible Score Range)	*n*	%
Depression (0–42)		
Normal-Mild (0–13)	10	41.6
Moderate (14–20)	9	37.5
Severe (≥21)	5	20.8
Anxiety (0–42)		
Normal-Mild (0–9)	13	54.2
Moderate (10–14)	4	16.7
Severe (≥15)	7	29.2
Stress (0–42)		
Normal-Mild (0–18)	13	54.2
Moderate (19–25)	3	12.5
Severe (≥26)	8	33.3

## Data Availability

The data presented in this study are available on request from the corresponding author.

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
