# Peer review of "Trans*Forming Access and Care in Rural Areas: A Community-Engaged Approach"

_ijerph, 2021, doi:10.3390/ijerph182312700_

Round 1

Reviewer 1 Report

This study explores an underdeveloped line of work due to the difficulties of access to this type of population. I think the study is very interesting and uses a mixed method that allows us to better understand the needs of transgender people. My suggestions for improving this article are as follows:

Method:

Why was an inclusion criterion having access to the internet? If you are exploring rural population, not allowing the participation of people without internet can skew the data collection. Include it in limitations.

Although they say that they focus on explaining the results of the 24 people who participate in the focus groups, then they include data from the online questionnaire that more people answer (line 146) .Table 1 only includes the demographic data of the participants in the focus group, but then the data of the people who answered the online survey are given. It should also include the general data or make another table. 

In Table 1 they include the White and Non-Hispanic categories, but it's actually the same, isn't it?

In Table 1 I see it unnecessary to include the transgender category, since it is the total of the rest of the categories. It would be more correct to put transwoman and transman.

The estimates of people in the focus groups do not match. It is said that there are 24, but there are 21 transgender people

Although they say that they focus on explaining the results of the 24 people who participate in the focus groups, then they include data from the online questionnaire that more people answer (line 146) .Table 1 only includes the demographic data of the participants in the focus group, but then the data of the people who answered the online survey are given. It should also include the general data or make another table.

Reviewer 2 Report

This manuscript reports on a small exploratory study about the healthcare experiences of transgender and gender diverse (TGD) individuals in West Virginia. A total of 24 TGD individuals and parents of TGD individuals responded to an online survey and participated in online focus groups. The authors provide a descriptive analysis about participants’ experiences with healthcare providers, and their perspectives towards healthcare services in WV. The results show that many had negative experiences and that much needs to be done to improve appropriate healthcare for TGD populations in rural areas.

I thought this article could fill an important gap in the extant literature, as I believe little has been published about the healthcare experiences of TGD populations in rural areas. The manuscript was overall well presented, though I had a few concerns and comments (described below) that I believe should be addressed before publication.

  1. I thought some details were lacking about how the quantitative results were obtained. For example, the “Perspectives on Providers” section provided a lot of details about how questions were asked, but this was not the case for the other subsections. On line 195, the authors write “participants reported that transportation represented a significant barrier in seeking care,” but was there a question asking participants how important transportation was as a barrier, or is that inferred from the other data presented in the paragraph? On lines 217–220, the authors write, “One participant (4.2%) is the temporary guardian of a minor and the minor wishes to have gender-affirming treatment but because the biological parents disallow it, the judge had not granted that authority to the temporary guardian.” This result does not seem quantitative; it looks like a response to an open-ended question. Those are only some examples, but there are other parts where I felt unclear regarding what was asked to participants. I think it will be easier to understand and appreciate the results if the authors are more transparent about what questions were used to obtain the data.
  2. I think the authors should say a more about the choice of having parents of TGD minors respond to the questions for them and how that choice was made. If available, I think the manuscript should provide details whether the minor participant and parent did the online questionnaire and/or focus group together, or whether the parent did everything without the minor participant. Although there are some questions that parents could answer for their child (e.g., demographics), it’s unclear that they could do so accurately for others. For instance, the validity of the DASS-21 might be questionable when someone else answers the questions for the participant (which should be included in the limitations). In the focus group questionnaire, some questions had an alternative phrasing for parents (e.g., Did you seek out specific providers to meet your child healthcare needs as a trans or gender diverse person?). In these cases, I don’t think we can say that parents are answering for the minor participants. The question is addressed directly to them, so the parents are actual participants in the study.
  3. Related to the above point, I thought that the basic description of the sample should be clearer throughout the paper (especially abstract and methods). It might be worth saying upfront that the participants are TGD individuals and parents of TGD minors. On line 290, there is a participant who is reporting something that happened to their transgender husband. The participant might also be transgender, but this quote raised the question whether cisgender partners of transgender individuals also participated in the study. I would clarify.
  4. The subheadings in the qualitative results are not themes; they are summaries of questions asked to participants (e.g., “Theme one: Supportive Healthcare Experiences” summarizes responses to “Can you think of a health care provider who did a great job providing care for you/your child?”). Themes would be topics that came up in participants’ answers and that cut across the data, across different questions. The results are well presented, but I would change the terminology of themes. I recommend this article for more on the topic of themes: Morse, J. M. (2008). Confusing Categories and Themes. Qualitative Health Research, 18(6), 727–728. https://doi.org/10.1177/1049732308314930
  5. I don’t believe that discussing the survey and focus group results together amounts to “integration” of the quantitative and qualitative data. Again, the results and discussion don’t really need to change, but I would remove the language about integration, as I don’t think both sets of qualitative and quantitative data were really integrated.
  6. Lines 97–106: I felt this paragraph could be brought down to one sentence, as the main topic of the article isn’t community engagement in research.
  7. I would add details about how many focus groups were conducted via Zoom, and about how many participants did the synchronous and asynchronous FGs.
  8. The tables could be improved by following guidelines from the APA or AMA. For instance, Table 1 is very hard to read because the demographic variables are all centered.
  9. On line 179, the text should specify what was the goal of bivariate analyses (e.g., outcome variable).
  10. It’s standard practice to report results in the past, but a lot of the results are written in present tense. I would also write about the study in the past, as the present tense implies that the study is still ongoing.
  11. In paragraph between lines 261–273, it seemed like the terms dependent and independent were used reversely (i.e., I feel like mental health is affected by access to gender-affirming healthcare, and thus that the DASS items should be the dependent variables).
  12. The writing felt overall good but could be improved with careful copyediting. Some sentences were awkward or clunky and could be revised. Just a few examples: Lines 66–68, this sentence includes “sexual risk behaviors” as a “healthcare need”: “When comparing rural and urban transgender healthcare needs, rural transgender persons have significantly higher needs in mental health, substance use, and sexual risk behaviors when compared to their urban counterparts [17].” Lines 89–91, too much in one sentence: “This mixed methods study, approved by the first author’s institutional review board for human subjects, examined the research question: what do TGD people need from medical providers in order to get their healthcare needs met in West Virginia?”
  13. In the discussion, I wondered if the authors could do more to compare/contrast their findings with related studies/data. For instance, lines 356–363, do the mental health outcomes differ from other studies with transgender individuals, or are there data about mental health in WV or rural areas (even if not among transgender populations) to contrast them to?
  14. As mentioned above, I think it’s important to mention that parents answered questions for minor participants as a limitation. An important limitation is also that results are self-reported and that there can be recall error.

Round 2

Reviewer 2 Report

I read the responses from the authors and the revised manuscript. I believe my comments have been appropriately addressed. I only saw one type on line 183 ("cae") but I recommend careful proofreading prior to publishing. I otherwise recommend this manuscript for publication in its current form.